# Use of a Composed Simulator by Veterinarian Non-Experts in Minimally Invasive Surgery for Training and Acquisition of Surgical Skills for Laparoscopic Ovariectomy in Dogs

**DOI:** 10.3390/ani13172698

**Published:** 2023-08-23

**Authors:** Belén Andrade-Espinoza, Carlos Oviedo-Peñata, Juan G. Maldonado-Estrada

**Affiliations:** 1OHVRI-Research Group, College of Veterinary Medicine, Faculty of Agrarian Sciences, University of Antioquia, Medellín 050010, Colombia; mabeanes@hotmail.com (B.A.-E.); juan.maldonado@udea.edu.co (J.G.M.-E.); 2Master of Science Program in Canine and Feline Internal Medicine, University of Cuenca, Cuenca 010107, Ecuador; 3Tropical Animal Production Research Group, Faculty of Veterinary Medicine and Zootechny, University of Cordoba, Monteria 230002, Colombia

**Keywords:** minimally invasive surgical procedures, models, ovariectomy, psychomotor performance, simulation training, spaying, training programs

## Abstract

**Simple Summary:**

In veterinary surgery, there is a growing demand for devices and curricula for laparoscopic surgery training which requires more realistic and low-cost training plans and simulators. There is still a lack of didactic training models providing a controlled and safe environment for the acquisition of advanced skills for specific surgical techniques. This work aims to evaluate the acquisition of advanced surgical skills training laparoscopic ovariectomy (LOE) using an ergonomic simulator obtained from a canine abdomen combined with real ovarian and uterine tissues freshly reconstituted from female reproductive tracts. All participants were evaluated using quantitative metrics and objective rating scales which resulted in significant improvements in surgical skill after training. We conclude that the proposed training curriculum and simulation device were appropriate for the acquisition of laparoscopic skills for simulated ovariectomy in female dogs. Training in ergonomic models of the canine abdomen combined with fresh reconstituted tissues improves surgical skills for LOE.

**Abstract:**

This study aims to assess the acquisition of surgical skills for laparoscopic ovariectomy (LOE) in dogs by veterinary surgeons with no experience in minimally invasive surgery using the CALMA Veterinary Lap-trainer simulator (CVLTS) in an experimental and analytical setting. Veterinary surgeons with no experience in minimally invasive surgery (MIS) (experimental, n = 5), and MIS experts (experts, n = 3) were evaluated. Experimental and expert group participants watched an instructional video (initial time) before practicing the LOE on uterine tissues and ovaries freshly reconstituted after elective ovariohysterectomy (initial time evaluation). Then, the experimental group practiced five training sessions on the composite simulator with permanent feedback and then performed the LOE again (final time evaluation). Surgical performances in initial and final evaluations were video recorded and further evaluated by three external MIS experts using Global objective assessment of laparoscopic skills (GOALS) and LOE-specific rating scales (SRSs) in a double-blinded schedule. In addition, a hands movement assessment system (HMAS) attached to the back of the hands was used to quantitatively measure completion time, angularity, and movement smoothness. Data were analyzed with one-factor ANOVA and Tukey’s contrast test. No statistically significant differences were found between the novice group’s performance after training and the expert group’s performance according to the GOALS (*p* < 0.01) and SRS (*p* < 0.05) scores. Moreover, the novices had significantly improved time, number of movements, and angularity in the final time compared with the initial time (*p* < 0.05), with no significant differences compared to the expert group (*p* > 0.05). LOE training using a composed simulator resulted in significantly improved laparoscopic skills and time, number, and angularity of movements data, providing evidence of the usefulness and reliability of CVLTS in training LOE.

## 1. Introduction

The growing practice of laparoscopy and minimally invasive surgery (MIS) in small [1,2] and large animals [3,4] has uncovered two critical facts: first, MIS requires prior simulation using appropriate models for the acquisition of basic, advanced, and specific skills for each surgery; and second, even though there are simulators for deliberate practice [5], there are not enough devices on which to train and perform the minimum simulation required to acquire the skills necessary to perform a given surgery [6]. Training for the acquisition of basic and advanced skills needs reliable devices in which the skills required to complete the actual procedure can be transmitted to the operating room [7,8,9], reducing the risks for the real patient and a curriculum supporting the training schedule [9]. Particularly, the skills of conventional surgery do not apply to MIS, emphasizing the importance of deliberate and feedback training for the acquisition of MIS skills [7,10,11].

MIS harbors advantages and disadvantages. Compared to conventional surgery, the advantages of MIS include the reduction of surgical trauma, postoperative morbidity, and postoperative infection; short recovery time due to smaller incisions; and, consequently, less postoperative pain [1,5]. The disadvantages are the high cost of the equipment and a training curriculum using simulators that requires a prolonged learning scheme [12]. In addition, training veterinary surgeons in small and large animal MIS demands the development of suitable simulators to learn the surgical protocols before applying them to the real patient [13,14]. This fact has also been evidenced in conventional curricula in veterinary schools [15,16,17]. Although most MIS veterinary surgeons have trained in simulators designed for human MIS in the last two decades, the number of simulators for veterinary use is increasing, although not at the expected rate of development. Moreover, its development is a continuous challenge for the achievement of competency-based learning [17]. Training to develop surgical competencies to perform laparoscopic ovariohysterectomy in bitches has been reported on simulators designed for veterinary [13,16,18] and human laparoscopy [19]. Canine spaying and neutering are among the most important surgical competencies for veterinary medicine students and residents of veterinary surgery [13]. Authors agree that one of the most critical factors for learning surgical skills for MIS is deliberate practice and the number of repetitions: a training program of 12 simulation sessions resulted in mean suture times that did not vary significantly between surgeons, in addition to achieving an 85% success rate in the completion of the ovariohysterectomy procedure by resident laparoscopic surgeons [13]. Accordingly, laparoscopic ovariohysterectomy (LOE) requires a learning curve of 80 procedures to reduce intraoperative complication rates [20]. In more advanced ovariohysterectomy techniques, such as laparoendoscopic single-site ovariectomy (LESS), the learning curve for experienced laparoscopic surgeons requires eight repetitions [21]. To enhance the didactic potential of simulators, specific hardware/software designs have been developed for hand motion analysis and surgical dexterity assessment enabling effective assessment of hand dexterity during surgical simulations [22,23]. Our research group constructed, evaluated, and validated the usefulness of a canine ergonomic model for veterinary laparoscopic surgery training called CALMA Veterinary Lap-trainer simulator (CVLTS) [24]. The construction of the simulator was based on a mold of a large breed canine abdomen, in which the Objective structured assessment of technical skill (OSATS) and GOALS evaluation systems and a surgical-procedure-specific scale according to the surgical procedure evaluated were incorporated. These experiments were performed successively to evaluate and validate basic skills [25] and advanced skills in laparoscopic surgery [25]. Similarly, the main purpose was to evaluate the transmission of surgical skills with the total laparoscopic gastropexy model after completing the training in the simulator and evaluating the performance of trainees in real conditions in TLG in the porcine model in vivo (manuscript submitted). The present work aims to evaluate the usefulness of CVLTS for the training and acquisition of surgical skills in laparoscopic right ovariectomy (LROE). Because there is no composite model—for example, simulators combined with real tissues for LOE training—our working hypothesis proposes that the use of CVLTS for laparoscopic surgery training using freshly reconstituted uterine tracts enables novice practitioners to acquire LOE skills similar to expert MIS skills.

## 2. Materials and Methods

This study received approval from the Institutional Board of Subject Experimentation of the University of Antioquia. This work is part of four studies in which different veterinarians (general practitioners) with more than five years of experience in conventional surgery but no experience in minimally invasive surgery received a training schedule and were evaluated for the acquisition of laparoscopic skills. The sample size was defined by convenience because of the limited level of development of MIS in our city [26]. The experimental group consisted of veterinarians experienced in conventional (open) surgery but not in MIS (novices, n = 5) and was evaluated and compared to veterinarian surgeons with MIS experience (experts, n = 3). Through a survey at inclusion, we recorded demographic data including age, sex, dominant hand, and previous experience in conventional surgery, laparoscopy, and video games. Participants signed informed consents. No live animals were used in the study. As shown in Figure 1, the protocol for LOE was performed by an MIS expert and was video recorded and edited to provide the participants with the specific steps of the protocol using the CVLTS and freshly reconstituted uterine tracts (see below). At the initial time, each veterinarian in the expert and novice groups realized the surgical protocol by watching the video once and then performing the procedure. Each participant’s performance was video recorded for further analysis and assessment of the GOALS and specific skills by two external experts in MIS unrelated to the experimental phase (Figure 1). In addition, during their performance, the participants wore a hands movement assessment system (HMAS) attached to the back of the hands (Figure 2) to quantitatively measure completion time, angularity, and movement smoothness.

Right LOE training protocol was performed in the laboratory of simulation, College of Veterinary Medicine, University of Antioquia (Medellin, Colombia). Canine reproductive tract fresh tissue fragments including testicles, sperm cordons, uterus, ovaries, and ligaments released after a community-based spay–neuter program were used for total uterine tract and ovaries reconstitution (Figure 3A,B). Only the novice group performed five repetitions of each specific task including (i) exposure of the right ovary, (ii) division of the broad ligament of the uterus, (iii) first ligature of the right ovarian pedicle, (iv) second ligature of the right ovarian pedicle and cutting, and (v) broad ligament cutting. The expert group performed the surgical protocol only once during the initial time evaluation (Figure 1). After completing the training sessions, the novice group performed the LOE that was video recorded (final time). Video recordings and HMAS data were further evaluated by external experts as indicated for the initial time assessment (Figure 1). 

The CLVTS was used for training and evaluation of novices and experts while performing five tasks required for completing LOE [24]. The HMAS used incorporates sensors registering hand movements during the initial time, simulation exercises, and final time, during which data information was transmitted in real time to a computer [22,23]. The set of dexterity metrics included (i) time to complete each task; (ii) number of movements; (iii) smoothness in movements; and (iv) angular displacement. 

A structured curriculum consisting of five repetitions of five LOE-specific tasks was used for novice training. In the construction of the study plan, important aspects were considered to ensure the learning of the technique, such as the theoretical session, the deliberate practice, and the sequencing of the technique in simpler steps [27,28]. Participants received a theoretical lesson supported by video, where they were instructed to correct common mistakes when performing each of the tasks required to complete the LOE. GOALS scales evaluated: depth perception, bimanual dexterity, efficiency, tissue handling, and use of Instruments using a 5-point Likert scale. 

The LOE-specific tasks included (1) exposure of the right ovary (Figure 4A) performed by holding the needle holder in the dominant hand and grasping a Maryland-type driver in the non-dominant hand, the participant fixes the right ovary on the abdominal wall on the same side using silk # 2-0 with CC needle 30, 3/8 circle (Silk master medical, Corpaul, Medellin, Colombia); (2) division of the broad ligament of the uterus (Figure 4B) performed by holding Maryland-type dissectors in both hands, the participant makes a hole in the right broad ligament with a blunt opening wide enough to allow two equidistant ligatures to be made and cut between both ligatures; (3) first ligature of the right ovarian pedicle (Figure 4C) was performed by holding the needle holder in the dominant hand and a Maryland-type dissector in the non-dominant hand, a ligature is made using the horizontal C and the inverted C with 17cm length silk # 2-0, without a needle (Master medical silk, Corpaul, Medellin, Colombia) in the distal part (see above) of the ovarian pedicle (close to the ovary) according to the report by Szabo [29]; (4) second ligature of the right ovarian pedicle and cutting (Figure 4D,E) performed by the participant making a second knot in the same way as the first one but proximal to the ovarian pedicle and an equidistance cutting between both ligatures using Metzenbaum scissors in the dominant hand; and (5) broad ligament cutting (Figure 4F) performed by holding Metzenbaum scissors in the dominant hand and grasping a Maryland-type dissector in the non-dominant hand, the surgeon cut the broad ligament in the direction of the body of the uterus avoiding the mimicked right uterine artery (Figure 4G). Double ligation with Roeder 4s extracorporeal suture of the uterine body using the technique and cutting the uterine body (Figure 4H) with Metzenbaum scissors were trained but not evaluated. Participants received real-time feedback when necessary. No warm-up exercises were allowed before the initial evaluation. 

Once participants realized the protocol, they wore the HMAS sensors on each hand and started LOE only on the right side of the previously reconstituted uterine tract containing the ovarian burse (Figure 3B). The reconstituted tracts were prepared between one and three days before each performance, kept under refrigeration at 4 °C, and warmed at room temperature before use or used freshly after reconstitution the same day after recovering. The uterine tracts were placed into the simulation device model by clamping with plastic tweezers (Figure 4A,G,H). The video recording camera allowed the participant to visualize real-time images corresponding to their performance while recording all the procedures. The video recording corresponding to the initial evaluation for the experimental group and the single recording for the expert group were recorded and stored for further analysis. Video recording started when the participants of the experimental group and experts indicated they were ready (ports-in-hand), and both instruments were visible on the screen. The time expended for handling ports was not recorded.

After initial LOE performance and recording, the experimental group proceeded with five repetitions of each task at separate times. After completing the training protocol, novices performed the final LOE as indicated for the initial one. Video recordings of the trainees’ initial and final performances and the experts’ performances [30] were sent to external veterinarian surgeons (n = 2) with more than ten years of experience both as veterinary laparoscopic surgeons and as tutors of laparoscopic surgery training for GOALS and SRS evaluation, who then assessed performances in a double-blinded manner. Likert-type scales [31] were used, where a score from 1 to 5 was rated for each item, to achieve a maximum score of 25. Data were converted into Excel files for further statistical analysis. 

Novices’ and experts’ performances were assessed by external experts (as previously indicated) evaluating each corresponding video recording using the Global operative assessment of laparoscopic skills (GOALS) scale and specific ranking scale (SRS) for the LOE. HMAS data were evaluated by an independent statistics expert.

For the statistical analysis, quantitative data were assessed for assumptions of normality and homoscedasticity. Data from GOALS, SRS scales, and motion data were compared with the Mann–Whitney U test. The Wilcoxon test was used to compare pre- and post-training assessments. Interrater reliability was assessed with the Intraclass Correlation Coefficient (ICC) [32]. Statistical tests and graphs were run with the statistical environment R v 4.1.2 (2021) under the RStudio v 1.4.1717 (2021) platform.

## 3. Results

The experimental group comprised two women and three men, all performing conventional surgery, and no one reported having previous experience or training in MIS. All were right-handed with a mean of 28 years old and less than 5 years of professional experience in conventional surgery. The experts were three males with more than five years of expertise after graduation and more than two years of experience in laparoscopic veterinary surgery or previous training in MIS, and all were right-handed. 

Table 1 shows the GOALS and LOE-specific scores given by the experts, and the HMAS data obtained in the initial and final evaluations. The time to completion, number of movements, and angular displacement of the experimental group significantly improved in the final time for tasks one, three (*p* < 0.01), and five (*p* < 0.05) compared to the initial time (Figure 5). Ranking scales significantly improved in novices in the final time compared to the initial time with no statistically significant differences in the final time performance compared with the experts (*p* > 0.05) (Figure 6). Based on GOALS and SRS data surgical performance improved significantly (*p* < 0.05) between the initial and final assessments for the experimental group. Scores close to those executed by the experts were found when comparing the final assessment of the experimental group and the expert group (Table 1, Figure 7) (*p* > 0.05).

In the final time, novices achieved similar performances compared to experts (*p* > 0.05) for the variables time, number of movements, and angular displacement. In contrast, neither the metrics for tasks two or four nor the smoothness of movement in all tasks were significantly different between groups (*p* > 0.05) (Table 1, Figure 5) (*p* < 0.01, general scale, and *p* < 0.05, specific scale).

The time required to perform each task and to complete the procedure (completion time) is shown in Table 2. The completion time after finishing the training period significantly improved in the experimental group after the training process compared to the completion time before training (*p* < 0.01). This economy resulted in a 49.6%-time reduction. Similarly, the completion time of the experts was significantly lower compared to the trainees both before and after training. Three out of five specific tasks performed by the experimental group significantly improved after training (task one: the exposition of the right ovary; task three: right ovarian pedicle first ligation; and task five: transection of the broad uterine ligament) (Table 2).

**Figure 5 animals-13-02698-f005:**
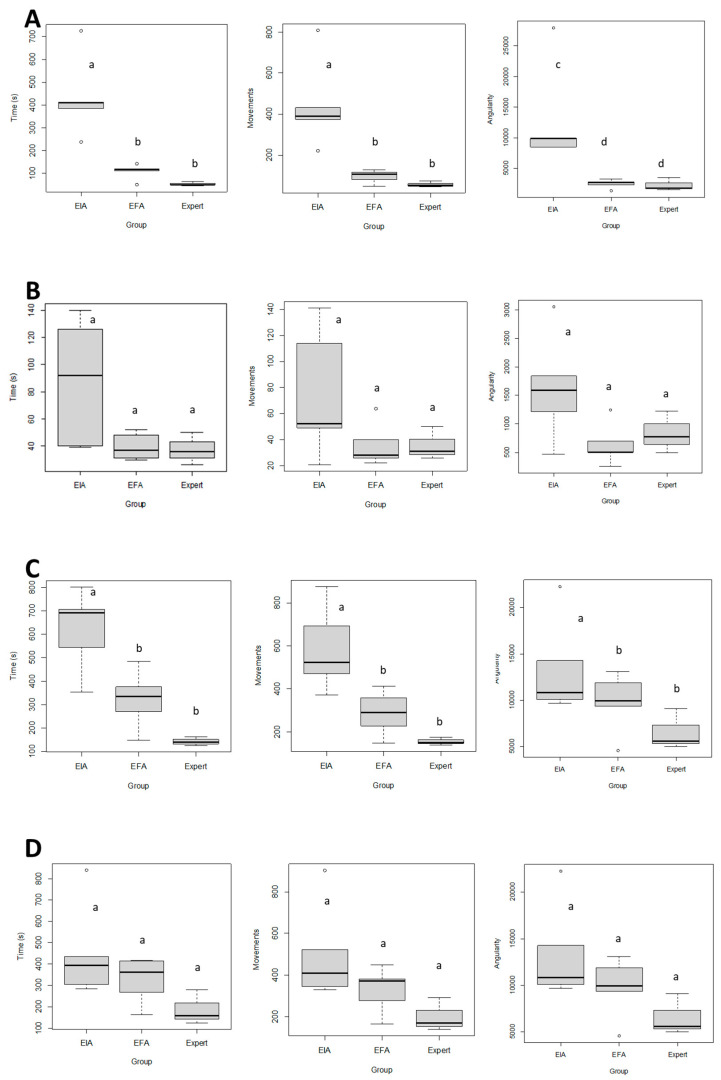
Motion evaluation of surgical skills. Box-and-whisker plot for the time, number of moves, and angularity of the five tasks. Data were analyzed with one-way ANOVA using a posteriori Tukey test. EIA: initial experimental assessment; EFA: a final experimental assessment. (**A**): task 1. (**B**): task 2. (**C**): task 3. (**D**): task 4. (**E**): task 5. Different letters between groups indicate statistically significant differences at *p* < 0.01 (a,b) and *p* < 0.05 (c,d). Y-axis: task number. X-top axis: time (**left column**). The number of movements (**central column**) and angular displacement (**right column**).

**Figure 6 animals-13-02698-f006:**
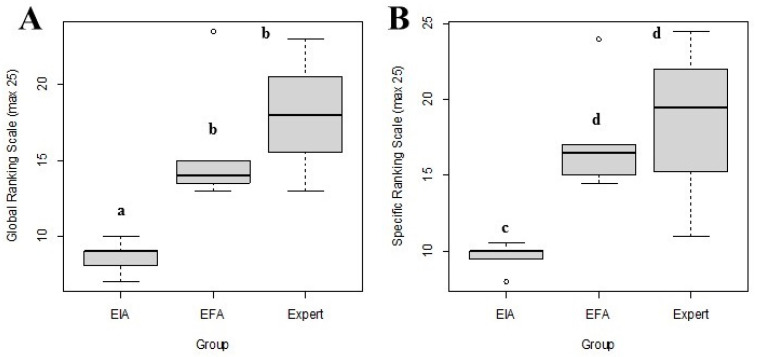
Box-and-whisker plot for the assessment of surgical performance with (**A**) general rating scale (GRS) and (**B**) LOE-specific rating scales (SRSs) when performing tasks during LOE. Data were analyzed with a posteriori Tukey test. EIA: initial experimental evaluation; EFA: final experimental evaluation; experts. Different letters between groups indicate statistically significant differences at *p* < 0.01 (a,b) and *p* < 0.05 (c,d).

**Figure 7 animals-13-02698-f007:**
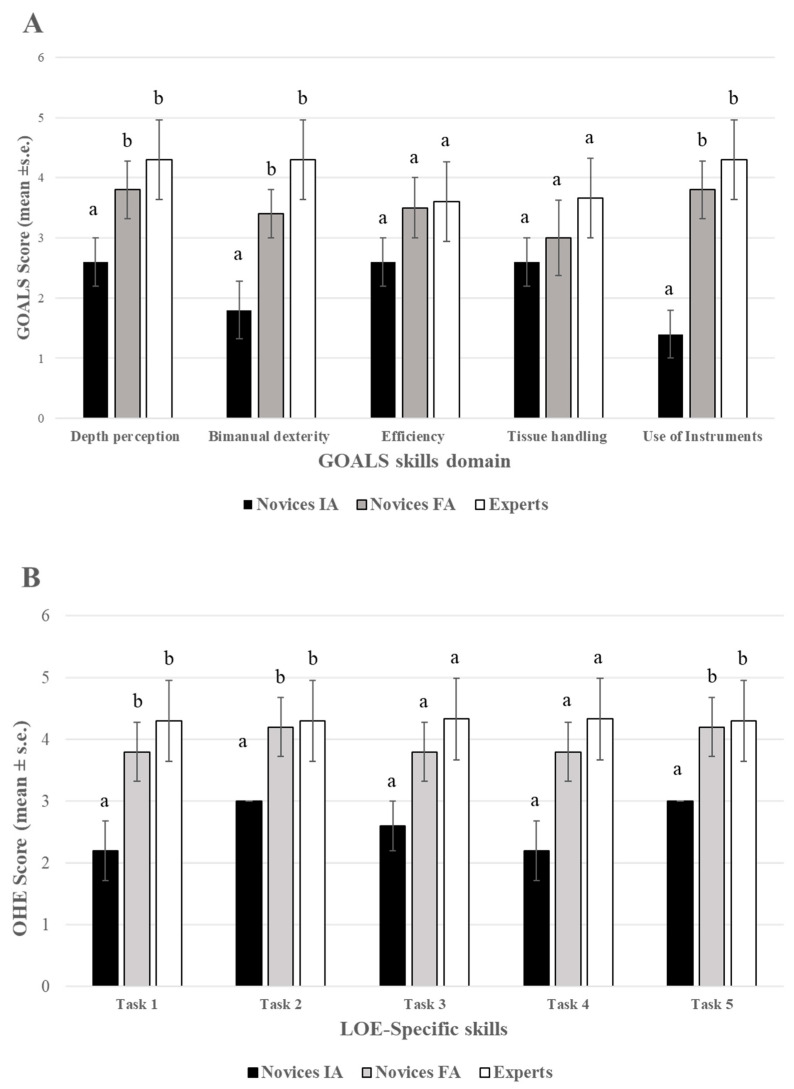
(**A**) Box-and-whisker plot for GOALS skills: (1) in-depth perception; (2) bimanual dexterity; (3) efficiency; (4) tissue handling; and (5) use of instruments. Data were analyzed with one-way ANOVA. Different letters within each skill mean a statistically significant difference (*p* < 0.05). (**B**) Box-and-whisker plot for LOE-specific skills: task one, exposition of the right ovary; task two, divulsion of the broad uterine ligament; task three, right ovarian pedicle first ligation; task four, right ovarian pedicle second ligation transection and cut; and task five, transection of the broad uterine ligament. Skills scores are shown for the initial assessment (black columns) and final assessment (gray columns) of the experimental group and experts (white columns). Data were analyzed with one-way ANOVA. Different letters within each task mean a statistically significant difference (*p* < 0.05).

The interrater correlation was poor (ICC < 0.50) for the GRS and SRS scales (Figure 6). Only moderate agreement was found for the items of bimanual ability and use of the instruments (Table 3). The score given to the experimental group by external tutor 2 was significantly lower (*p* < 0.01) for in-depth perception, tissue handling, and efficiency of the protocol compared to the score given by external tutor 1, although these differences disappeared in the final time (*p* > 0.05). Similarly, the scores for four out of the five LOE-specific skills were significantly lower as assessed by external tutor 2 compared to external tutor 1. No statistically significant differences were found between groups for tasks two and four (Figure 7).

## 4. Discussion

In this study, for the first time, a composed model for LOE in dogs was evaluated using two qualitative (GOALS and specific scores) measurements and quantitative HMAS measurements. Validation of the training curriculum is supported by the significantly improved skills of the experimental group after finishing the training protocol. The data also evidenced the lack of expertise in the novice group in the initial assessment and the appropriate inclusion of the expert group. External tutors and quantitative data could not differentiate the novices’ skills in the final time compared to the experts, providing evidence of the construct validity. The percentages of the GOALS and GRS scales achieved by the experts and the experimental group were 72% and 78% and 56% and 66%, respectively. However, we did not assess the significant transfer of these MIS skills to the real conditions of patients in the operative room.

In our work, the times were counted in the same way, which resulted in ligation times with a pedicle cutting of 11.63 min. The total completion time of the entire procedure was significantly reduced from 33.55 to 16.5 min in the experimental group in the final versus initial evaluation and compared to the experts (9.44 min) (Table 2). The experts’ time was twice that reported by Chen et al. performing simulated left LOE in a box model who reported 9.6 min for the novices and 9 min for the intermediate group, with a significant difference from the group of experts (4.8 min) [18]. Completion times of 28.8 min (range 20.5–39 min) were reported for LOE on a real patient [19]. Other authors reported an 85% (11/13) success rate for the laparoscopic surgeries performed by the experimental group, with a median total ligation time of 33 min (range 17 to 57) for both ovaries [13]. In the report by Tapia-Araya et al. (2015), who evaluated comparisons between one-port and two-port training of LOE in actual patients, the authors reported an average completion time of 36.6 and 32.0 min, respectively, in agreement with our finding regarding the novices in the initial assessment [33]. Differences in our novices’ completion times after final training and the experts’ times probably occurred because our groups did not perform the complete procedure of LOE. Accordingly, the steps not performed included port placement, change of position from the right to the left ovary, tissue removal, and pneumoperitoneum release [18]. In our study, time was recorded once the subjects were firsthand at the ports. In our study the total time of novices after training and that of experts is within the range reported by Bydzovsky et al. for laparoscopic-assisted EHO in dogs [34]. 

The experimental group in our study significantly improved bimanual dexterity and use of instruments for three out of five tasks (Figure 7). No efficiency or tissue handling improvement was found for novices in the final evaluation compared to the initial evaluation, which could mean these are skills that are not specific to laparoscopic surgery.

In our study, no agreement was established between the evaluations of the two double-blind external experts (Table 3). We noticed that the low scores corresponded to an evaluator who only used the odd numbers of the Likert scale (1, 3, and 5) while the other used all five points of the scale. The lack of external information, because the video recordings did not include information outside the simulator environment, could have influenced the experts’ scores. Scott et al. conducted a study where they compared the global evaluation made from edited video recordings versus direct observation scores and found no correlation with direct observation, the latter being better for evaluating simulated training [30]. In our work, the differences between the external evaluators occurred for the data from the initial time of evaluation of the novices, which can be attributed to the fact that direct observation is better for evaluating newbies in a simulated training program because an overall evaluation through video recording does not correlate many times with direct observation and there is little reliability among the evaluators [35]. It is intriguing that as the performance of the experimental subjects improved, the discrepancy between the external evaluators also increased, but we have no explanation for this finding.

In our work, no statistical differences were found for task two (divulsion of the broad uterine ligament) or task four (right ovarian pedicle second ligation transection and cutting) metrics. One explanation is the simplicity of task two, which did not imply a greater effort for making the hole through the avascular and more transparent portion of the broad ligament before the ligatures. Task four corresponded to intracorporeal suture using a 17cm length # 2-0 silk suture without a needle and cutting the ovarian pedicle between the two finished ligatures. The lack of a statistically significant difference is due to the repeated exercise regarding task three (first ligature), as evidenced by the improved time, number of movements, and angularity (Table 1), supporting the concept that a single repetition positively influences the acquisition of surgical skills.

In the study by Au Yong et al., the authors suggested that simulation could improve veterinary students’ performance in an actual patient [36]. The significant transfer of surgical skills to a real patient was beyond the objectives of our study. However, our model proved that it could help improve specific laparoscopic surgical skills for LOE in dogs. Accordingly, the novices performed five consecutive repetitions of each task for three weeks, in agreement with the report by Freeman et al. who evaluated veterinary medicine students’ laparoscopic skills, highlighting the importance of repetition in the acquisition of surgical skills [37]. 

The GOALS scale results had higher variability than the specific scale, possibly because the training was based on a specific surgical technique such as LOE and not on basic training tasks. Accordingly, the overall performance scales only detect differences during training that include maneuvers alluding to a surgical act because the training based on basic tasks does not improve the scores on the performance scales [38].

In the training sessions where deliberate practice was applied, the participants quickly improved their performance on the scales and the Mocap device, demonstrating that the feedback was conducted correctly. Therefore, deliberate practice as a teaching strategy improves laparoscopic surgical skills [38]. In the final evaluation, the novice group’s evaluation was more homogeneous, showing significant improvements similar to the expert level. We suggest that a critical aspect of the acquisition of the surgical skills demonstrated in our work lies in the construction of the training program based on elements of the constructivist pedagogical approach, which conceives learning as the process of building new knowledge and attitudes based on existing ones in cooperation with classmate and teacher feedback. In addition, students have a huge responsibility to build their knowledge (skills) with a tutor’s help and repetitive practice [17,39,40].

The external characteristics of our simulator facilitate adequate positioning for both laparoscopic ovariectomy and laparoscopic right-sided ovariohysterectomy with linear port placement (Figure 3). Likewise, the fact that it is a composite simulator enables the use of fresh tissues that improve the haptic feedback and the replication of the anatomical model once the fragments of the reproductive tracts are available. In the market, simulators have been created for left ovariectomy using linear ports with good acceptance but with difficulties because simulators do not allow lateral decubitus tilt, haptic feedback due to the use of latex, or the realism of the procedure in general [18]. Likewise, in a study that evaluated a simulator for standing equine laparoscopic ovariectomy (SELO) [41], latex was also the object of complaint when performing amputation of the ovary and cutting the ovarian pedicle due to material texture. In our study, these inconveniences were partially solved by the uterus and ovarian sac reconstruction.

Difficulties have also been found in the simulation models related to their design, which does not adjust to the patient's position in real surgery; therefore, it has been suggested to develop simulators that allow a differentiated training for the handling of instruments in the horizontal and vertical planes [41,42]. Our proposal for LOE evaluation and training with the linear ports at the time of design and construction of the CVLTS was adjusted for the tilted position of a dag when undergoing LOHE or LOE.

In this study, the experimental and expert groups only simulated the suture and incision of the right ovary. No evaluation of the acquired laparoscopic skills by the experimental group was conducted in the operating room. One limitation of our model is that it requires fresh tissues for reconstituting the complete uterine tract, a fact that could limit its application in settings not closely related to spay–neuter programs, in addition to potential limitations for organic residue disposal. Although in our study there was no previous experience in basic skills in laparoscopic surgery, prior training in a validated laparoscopic basic skills curriculum could further guarantee success in an advanced training program such as the one we propose for laparoscopic LOE. Another limitation was not studying the transfer of the surgical skills that the simulator and curriculum could transfer to the operating room.

## 5. Conclusions

We provide evidence on the importance of practicing repeated laparoscopic surgical skills for LOE in dogs using the CVLTS and reconstituted reproductive tissues, which provide the apprentices with the natural texture and conditions of the uterine and ovarian tissues. This model also provides a more accurate ergonomic device for training these and other laparoscopic surgical skills in dogs as reported elsewhere [24,25]. Our study proposes an advanced training plan to develop the laparoscopic skills required to perform ovariohysterectomy in female dogs. The simulated laparoscopic ovariectomy proposed in our advanced training program is practical and feasible for replication as a didactic means for teaching LOHE and LOE, as it allows students to significantly improve their laparoscopic skills.

## Figures and Tables

**Figure 1 animals-13-02698-f001:**
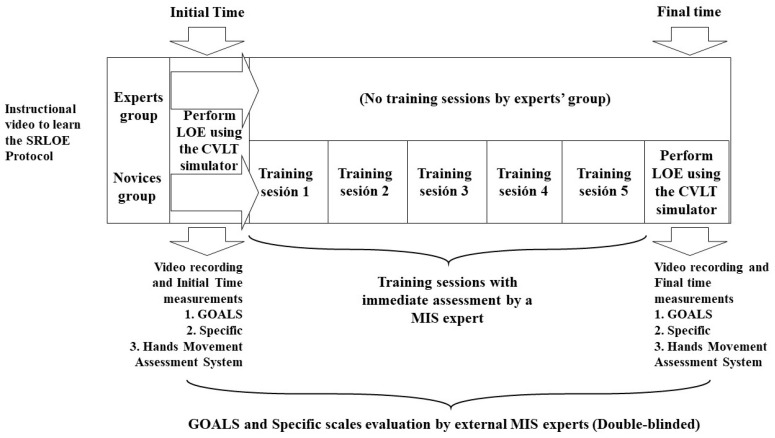
Summarized experimental model using the CALMA Veterinary Lap-trainer simulator (CVLTS). LOE—laparoscopic ovariectomy. GOALS—Global operative assessment of laparoscopic skills. SRS—specific ranking scale (for a given surgery, in this case, LOE).

**Figure 2 animals-13-02698-f002:**
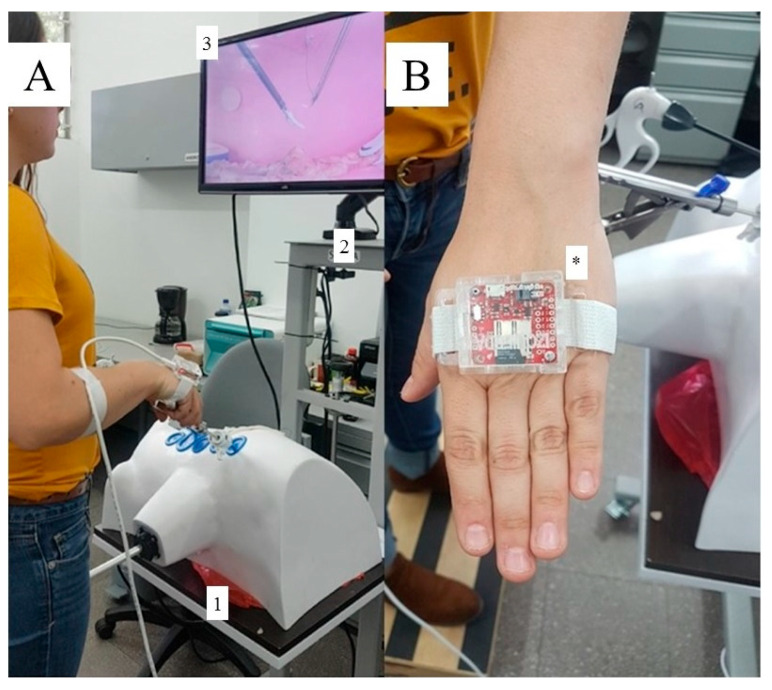
CALMA Veterinary Lap-trainer simulator (CVLTS). (**A**) Positioning of the veterinarian surgeon in front of CVLTS showing the ergonomic simulator (1), tower supporting the simulator (2), and screen connected to the video-recording camera located inside the simulator (3). (**B**) Positioning the hands movement assessment system (HMAS) on the veterinarian surgeon’s hands (asterisk).

**Figure 3 animals-13-02698-f003:**
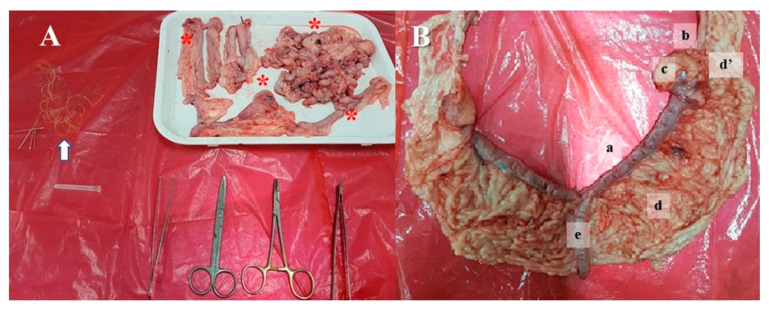
(**A**) Fresh tissues and materials from elective spaying and neutering (asterisks) outreach services, Municipality of the Medellin (Colombia); arrow: suture material. (**B**) The resulting reconstituted uterine tract and ovaries: (a) uterine horn, (b) suspensory ligament of the ovary, (c) ovarian bursa containing the ovary, (d) broad ligament of the uterus, (d′) mesometrium, and (e) uterine body and cervix.

**Figure 4 animals-13-02698-f004:**
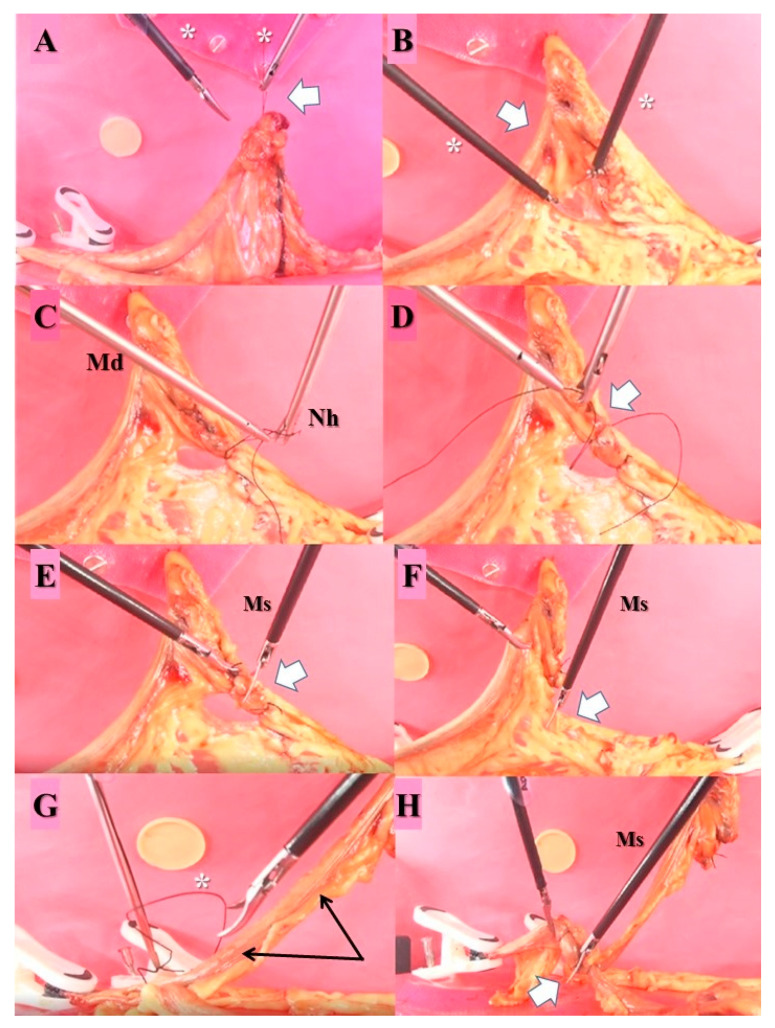
(**A**) Exposure of the ovary to the silicone patch (asterisks) mimicking the abdominal wall by passing a percutaneous needle (white arrow). (**B**) Divulsion of the broad ligament behind the anatomical area of the ovarian pedicle (white arrow) with Maryland forceps (asterisks). (**C**) First ligation of the ovarian pedicle holding the needle holder (Nh) in the dominant hand and a Maryland-type dissector (Md) in the non-dominant one. (**D**) Second ligation of the ovarian pedicle dorsal to the first one (white arrow). (**E**) Cutting the ovarian pedicle with Metzenbaum scissors (Ms) (white arrow). (**F**) Broad ligament cutting (white arrow) using Metzenbaum scissors (Ms). (**G**) Double ligation with Roeder 4s extracorporeal suture (asterisk) of the uterine body avoiding the mimicked right uterine artery (black arrows). (**H**) Cutting of the uterine body with Metzenbaum scissors (white arrow).

**Table 1 animals-13-02698-t001:** Metrics of tasks obtained with the hands movement assessment system in the advanced laparoscopic training program for LOE.

Items		Experimental Initial Assessment	Experimental Final Initial Assessment	Expert	*p*
**Task one**	Time (s)	411 (311.5–568.5) ^a^	117 (81.5–131.5) ^b^	51 (48–66) ^b^	0.002
	Movement (n)	391 (298.5–620.5) ^a^	106 (65–122) ^b^	52 (47–72) ^b^	0.003
	Smoothness in movements (degrees)	0.33 (0.235–0.44)	0.26 (0.245–0.285)	0.31 (0.31–0.36)	0.312
	Angularity (degrees)	9861.1 (8472.285–18,925.24) ^a^	2700.68 (1809.64–3032.395) ^b^	1758.54 (1558.44–3544.75) ^b^	0.020
**Task two**	Time (s)	92 (39.5–133)	37 (30.5–50)	36 (26–50)	0.059
	Movement (n)	52 (35–127.5)	28 (24–52)	31 (26–50)	0.174
	Smoothness in movements (degrees)	0.26 (0.2–0.425)	0.22 (0.185–0.245)	0.23 (0.22–0.27)	0.362
	Angularity (degrees)	1586.39 (841.41–2446.535)	503.56 (374.26–970.475)	779.83 (497.38–1218.84)	0.093
**Task three**	Time (s)	692 (448–754) ^a^	336 (208.5–429) ^b^	140 (126–162) ^b^	0.002
	Movement (n)	523 (422–784.5) ^a^	289 (188–384.5) ^b^	151 (140–175) ^b^	0.004
	Smoothness in movements (degrees)	0.31 (0.26–0.44)	0.29 (0.28–0.3)	0.33 (0.31–0.33)	0.556
	Angularity (degrees)	15,692.84 (11,714.06–19,720.33) ^a^	8719.37 (4792.395–11,426.67) ^b^	4804.71 (4288.24–5294.87) ^b^	0.005
**Task four**	Time (s)	394 (294.5–638)	362 (214–416.5)	157 (124–279)	0.129
	Movement (n)	410 (336–711.5)	372 (222.5–414.5)	170 (140–292)	0.086
	Smoothness in movements (degrees)	0.32 (0.265–0.45)	0.3 (0.29–0.31)	0.31 (0.3–0.35)	0.653
	Angularity (degrees)	10,828.43 (9901.155–18,235.1)	9903.98 (6973.965–12,475.27)	5577.51 (5038.23–9080.52)	0.106
**Task five**	Time (s)	407 (283–455) ^a^	139 (106–165.5) ^b^	183 (149–263) ^b^	0.002
	Movement (n)	375 (187–449) ^a^	107 (84.5–127) ^b^	214 (154–258) ^a^	0.011
	Smoothness in movements (degrees)	0.27 (0.22–0.54)	0.25 (0.235–0.295)	0.29 (0.27–0.3)	0.392
	Angularity (degrees)	11,825.17 (4006.35–13,704.56) ^a^	2193.09 (1767.04–2873.15) ^b^	5544.27 (3812.95–6283.79) ^b^	0.024
**GOALS GRS**		9 (7.5–9.5) ^a^	14 (13.25–19.25) ^b^	18 (13–23) ^b^	0.009
**SRS**		10 (8.75–10.25) ^a^	16.5 (14.75–20.5) ^b^	19.5 (11–24.5) ^b^	0.015

Values are presented as medians. Q1 and Q3 correspond to the lower and upper edges. The superscripts indicate Tukey’s a posteriori test; the same superscripts show that there are no statistical differences between groups, and the different superscripts show that there are statistically significant differences (α = 0.05). The tasks included: (1) exposure of the right ovary, (2) division of the broad ligament of the uterus, (3) first ligature of the right ovarian pedicle, (4) second ligature of the right ovarian pedicle and cut, and (5) cutting broad ligament. HMAS metrics: operating time (seconds), total movements (n), total smoothness in movements (n), total angular displacement (degrees).

**Table 2 animals-13-02698-t002:** Partial and completion time for LOE tasks performed by the novices before and after the training compared to the experts.

		Time Required for Achieving the Tasks (min)					
Group	Assessment	Task One	Task Two	Task Three	Task Four	Task Five	Total
Experimental	Initial	6.85 ^a^	1.53 ^a^	11.53 ^a^	6.56 ^a^	6.78 ^a^	33.25 ^c^
	Final	1.95 ^b^	0.61 ^a^	5.6 ^b^	6.03 ^a^	2.31 ^b^	16.50 ^d^
Experts		0.85 *	0.60	2.33	2.61	3.05	9.44

Activity with different letters (^a,b^) means statistically significant differences (*p* < 0.05, Student’s *t* test). ^c,d^ means statistically significant differences (*p* < 0.01, Student’s *t* test). Task one: the exposition of the right ovary; task two: divulsion of the broad uterine ligament; task three: right ovarian pedicle first ligation; task four: right ovarian pedicle second ligation transection and cutting; and task five: transection of the broad uterine ligament. Asterisks within a column mean statistically significant differences between novices at time zero and experts (*: *p* < 0.05, Student’s *t* test).

**Table 3 animals-13-02698-t003:** Interrater reliability for the GOALS scale.

GOALS General Ranking Scale	Depth Perception	Bimanual Skills	Efficiency	Tissue Management	Use of Instruments	Total GRS
ICC	0.172	0.611	−0.0138	−0.04	0.59	0.435
*p*	0.271	0.0083	0.516	0.552	0.0109	0.0544
Specific Ranking Scale	Task 1	Task 2	Task 3	Task 4	Task 5	Total SRS
ICC	0.408	0.296	0.248	0.217	0.402	0.353
*p*	0.0675	0.145	0.189	0.221	0.0709	0.101

Poor correlation: <0.5, Moderate: 0.51–0.75, Good: 0.76–0.9, Excellent: >0.9. (Koo and Li, 2016) [32].

## Data Availability

All data relating to the study are recorded in excel files, statistic data, and video records of each participant are available under request.

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
