# Peer review of "Use of a Composed Simulator by Veterinarian Non-Experts in Minimally Invasive Surgery for Training and Acquisition of Surgical Skills for Laparoscopic Ovariectomy in Dogs"

_animals, 2023, doi:10.3390/ani13172698_

Round 1

Reviewer 1 Report

Dear authors,

Thank you very much for your submission. I enjoyed reading this manuscript. Objective evaluation of the effectiveness of minimally invasive surgical training is critical for future veterinary education. Although this manuscript is very interesting, the materials and methods are complex, and it is difficult to know what training was evaluated and how. It is unclear what the authors want to convey to the reader. It would be helpful if you could be clearer about what you want to convey to the reader from these examples. In addition, the abbreviations are inadequate, the proofreading of the text is difficult to understand, and the manuscript does not stand up as a manuscript. Unfortunately, it is not suitable for submission in its current state.

There are fair comments. However, these are good examples and should not be taken negatively, as I think some changes would contribute to this literature group. It is recognized that it is difficult to quantify or qualify the effectiveness of education. Please review the submission guidelines and resubmit after making any corrections.

Title

Is CVLTS a common acronym? Acronyms in the first appearance should be spelled out. The same applies below.

Simple summary

Line 22: Please consider spelling out LOE.

Abstract

Just reading the abstract, it is unclear what data was accumulated and how, and what it means. How about describing that in detail? If there is a character limit, please delete the statistical processing section.

Line 24: Please consider spelling out CVLTS.

Line 25: I think the sentence should be established ‘Experimental analytical’.

Line 29: ‘GOALS and specific (SRS) LOE scales’  What does this mean?

Line 30: ‘a Hands Movement Assessment System (HMAS) ‘  Is it not in lower case?

Line 35-36:’without 35 significant differences with the expert’s group (P>0.05).’  What does this mean?

Keywords

Is the word Castration not commonly used in males? neutralization, spay…

‘Minimally Invasive Surgical Procedures’  Lower-case except at the beginning of a line

Introduction

Line 47-49: Consider providing references.

Line 51:  Reference 1 could not be verified. Please list other references.。

Line 52:  Move [(2)] to the appropriate place.

Line 59 and 66: ovariohysterectomyovariectomy?

Please consider spelling out.

line 66: OE

line 75: CALMA

line 77: OSATS and GOALS

Line 83: TLG

Materials and Methods

It is difficult to understand. I would like to see a concise, orderly description of what was evaluated and how. Please use paragraphs and headings to clarify each evaluation item and how it was evaluated. One way to make it easier to understand is to show it in a flowchart.

Please consider unifying the notation of either Experimental or Novices.

Fig 2

What do these mean? tower’ and Iglove?

Results and Discussion

I do not fully understand Materials and Methods, so I will refrain from commenting on it.

Author Response

Dear

Reviewer 1.

We appreciate the comments and suggestions. We send attached the corrections 

Sincerely, 

C. Oviedo  

Reviewer 2 Report

The authors report the use of a simulator with anatomical pieces for training in laparoscopic ovariectomy. While the study is interesting and the methodology is correct, I have some concerns about the basis of the study.

Why did you compose groups with non-homogeneous numbers of surgeons?

Do you think that such small numbers may have influenced the statistical results, i.e. that in a more representative sample the statistical results may be different?

Of course, the lack of clinical application of the simulator cannot confirm the validity of the instrument or whether it is better than others, even if you have declared that it was not the purpose of your paper.

Perhaps, in my opinion, it may be convenient to clarify that we are dealing with "first experiences ...."

Author Response

Dear 

Reviewer 2.

We appreciate the comments and suggestions. We send attached the corrections in the letter to the editor

Sincerely

C. Oviedo

Round 2

Reviewer 2 Report

I acknowledge the corrections of the authors.

Author Response

.
